# Survey of Physicians and Healers Using Amygdalin to Treat Cancer Patients

**DOI:** 10.3390/nu16132068

**Published:** 2024-06-28

**Authors:** Sascha D. Markowitsch, Sali Binali, Jochen Rutz, Felix K.-H. Chun, Axel Haferkamp, Igor Tsaur, Eva Juengel, Nikita D. Fischer, Anita Thomas, Roman A. Blaheta

**Affiliations:** 1Deparment of Urology and Pediatric Urology, University Medical Center Mainz, 55131 Mainz, Germany; sascha.markowitsch@unimedizin-mainz.de (S.D.M.); jochen.rutz@unimedizin-mainz.de (J.R.); axel.haferkamp@unimedizin-mainz.de (A.H.); igor.tsaur@med.uni-tuebingen.de (I.T.); eva.juengel@unimedizin-mainz.de (E.J.); nikita.fischer@unimedizin-mainz.de (N.D.F.); anita.thomas@unimedizin-mainz.de (A.T.); 2Department of Urology, Goethe-University, 60590 Frankfurt am Main, Germany; seese121314@gmail.com (S.B.); felix.chun@ukffm.de (F.K.-H.C.)

**Keywords:** amygdalin, survey, cyanide intoxication, treatment regimen, patient communication

## Abstract

Amygdalin is purported to exhibit anti-cancer properties when hydrolyzed to hydrogen cyanide (HCN). However, knowledge about amygdalin efficacy is limited. A questionnaire evaluating the efficacy, treatment, and dosing protocols, reasons for use, HCN levels, and toxicity was distributed to physicians and healers in Germany, providing amygdalin as an anti-cancer drug. Physicians (20) and healers (18) provided amygdalin over 8 (average) years to nearly 80 annually treated patients/providers. Information about amygdalin was predominantly obtained from colleagues (55%). Amygdalin was administered both intravenously (100%) and orally (32%). Intravenous application was considered to maximally delay disease progression (90%) and relieve symptoms (55%). Dosing was based on recommendations from colleagues (71%) or personal experience (47%). If limited success became apparent after an initial 3g/infusion, infusions were increased to 27g/infusion. Treatment response was primarily monitored with established (26%) and non-established tumor markers (19%). 90% did not monitor HCN levels. Negative effects were restricted to a few dizzy spells and nausea. Only 58% were willing to participate in clinical trials or contribute data for analysis (34%). Amygdalin infusions are commonly administered by healers and physicians with few side effects. The absence of standardized treatment calls for guidelines. Since intravenous application bypasses metabolization, re-evaluation of its mode of action is required.

## 1. Introduction

Cancer, the leading cause of global death, accounted for approximately 19 million new cases and 10 million deaths in 2020. Fueled by population growth and aging, it is estimated that the number of cancer cases will increase to over 28 million by 2040 [1].

Conventional tumor treatment includes surgical tumor resection, chemotherapy, and radiotherapy, either as single treatments or in combination. Despite significant therapeutic progress due to the recent approval of tyrosine kinase and immune checkpoint inhibitors, severe side effects and resistance development still prevent optimal drug efficacy. Dissatisfaction with conventional cancer protocols and the desire to activate the immune system in pursuit of a cure drives many cancer patients to seek complementary and alternative medicine (CAM) options. The term CAM has not been exactly defined but includes medical products and practices not included in standard medical care. Studies conducted in France and Poland have indicated that 85% of tumor patients integrate CAM into their treatment plans [2,3]. A regional survey in Sweden documented that more than 50% of cancer patients use CAM [4], and an evaluation in Germany estimated that 50–90% of patients apply CAM [5,6,7,8].

Plant extracts or isolated plant-derived compounds are widely utilized CAM options. However, the popularity of herbal drugs is not met with accordant knowledge of their mode of action, and evidence-based clinical trials are sparse. Physicians and healers thus offer natural phytochemicals as potent remedies, although their benefits in terms of tumor suppression or immune augmentation remain unclear. Cancer patients themselves often seek advice from sources of unknown quality without critical evaluation [9]. The discrepancy between the use of a phytodrug and knowledge about its mode of action is particularly evident for the cyanogenic diglucoside amygdalin (D-mandelonitrile-β-D-glucosido-6-β-D-glucoside), highly concentrated in fruit kernels from Rosaceae species such as peach and apricot. Amygdalin gained significant attention in the United States in the 1970s, with an estimated 70,000 annually treated cancer patients [10].

The therapeutic use of amygdalin includes both oral and intravenous (i.v.) applications. Orally consumed amygdalin undergoes metabolism in the gastrointestinal tract, producing prunasin, mandelonitrile, glucose, benzaldehyde, and cyanide (HCN) through enzymatic β-glucosidase activity. HCN is subsequently detoxified through enzymatic conversion by rhodanase to thiocyanate (thiosulfate sulfurtransferase) [11]. In contrast, i.v. administered amygdalin is excreted primarily unchanged, bypassing enzymatic degradation [12].

The relevance of amygdalin as an anti-tumor drug is ambiguous. Under the presumption that tumor cells contain high amounts of β-glucosidase but reduced amounts of rhodanase compared to normal cells, proponents of amygdalin therapy claim that amygdalin selectively kills tumor cells by leading to an accumulation of HCN in these particular cells [11]. Opponents of amygdalin therapy, however, warn that amygdalin is ineffective and HCN might systemically accumulate in the body with severe cyanide poisoning. Due to the potential risk of HCN intoxication, the sale of amygdalin for medical use has been forbidden in the United States and Europe (with few exceptions) [13,14]. Nevertheless, amygdalin is still administered with unknown prevalence as an anticancer drug.

Amygdalin has regained increasing attention over the last decade. However, although discussion about amygdalin’s clinical relevance has resurfaced, critical analysis is hindered by a lack of detail in medical practice. Aiming to provide updated insight into treatment concepts, a survey was distributed to physicians and healers who offer amygdalin treatment in Germany. Reasons for administration, dosing schedules, patient numbers, scientific training, efficacy assessment, incidence of toxicity, and degree of patient monitoring, including HCN detection, were surveyed.

## 2. Materials and Methods

### 2.1. Study Questionnaire

A questionnaire consisting of 43 questions, categorized into four sections was conceived. The four sections were as follows:(1)Demographic data, including age, gender, and education.(2)Use of amygdalin, experience, and assessment of its therapeutic potential.(3)Source of information on amygdalin and patient communication.(4)Therapeutic strategy, evaluation of response, and assessment of toxicity.

Some questions allowed only a single response, requiring participants to choose one option from several predetermined answers. Other questions provided the opportunity for multiple responses from a set of predetermined options. A specific set of questions allowed ranking from 0 to 10 (0 = no information provided, 10 = maximum information provided). Another set allowed participants to respond with “Yes” or “No”. Participants were also encouraged to provide additional free-text comments when “other” was indicated. All data were collected anonymously. The questionnaire did not solicit personal information (such as name and address) that could identify the participant. The survey period extended from August 2020 to November 2021.

### 2.2. Study Participants

Our search identified physicians and alternative practitioners (healers) in Germany whose websites indicated they administered amygdalin to treat cancer patients. They were subsequently informed about our study via email or telephone. All individuals were invited to participate in the online survey with assurance that participation was voluntary and anonymous. Of the 76 providers contacted, 60 expressed interest and agreed to participate. Questionnaires were sent to them by mail, along with a prepaid envelope for anonymous return. Of the 60 participants, 38 completed and returned the questionnaires correctly by mail, maintaining anonymity.

### 2.3. Ethics

According to the guidelines of the ethics committee at Goethe University Hospital Frankfurt, ethical approval for this anonymous survey was deemed unnecessary.

### 2.4. Statistical Analysis

Data were analyzed using BiAS 10.04. (BiAS for Windows, EPSiLON, Frankfurt, Germany). Mean value and standard deviation were calculated. Correlation analysis was conducted using Software R (Version 3.1.4, R Foundation for Statistical Computing, Wien, Austria). A *p*-value below 0.05 was considered significant.

## 3. Results

### 3.1. Demographic Data and Education

Thirty-eight participants (18 healers and 20 physicians) who completed the questionnaire were enrolled in the study. The mean age of the study cohorts was 58.3 ± 8.1 years (min: 37, max: 81 years), with no significant differences between the groups (healers: 57.4 ± 8.7 years, physicians: 59.0 ± 7.5 years). Twenty-six participants (68.4%) were male, and 12 were female (31.6%). Gender distribution among healers was balanced (9 male versus 9 female). In contrast, the physicians’ group exhibited significant gender disparity, with 17 (85%) being male and only 3 (15%) being female (*p* = 0.02) (Table 1).

Seven physicians were general practitioners, and 13 had completed medical specialist training. Both physicians and healers had obtained additional education related to complementary tumor treatment and patient care. However, more physicians had gained knowledge in acupuncture (*p* = 0.01), homeopathy (*p* = 0.01), and/or naturopathy (*p* = 0.03) (Table 2) compared to healers.

### 3.2. Use of Amygdalin

All participants had administered amygdalin for 8.8 ± 5.5 years with no significant differences between physicians and healers. Overall, 18.6 ± 48.2 tumor patients were treated quarterly with amygdalin. Although the number of patients visiting healers exceeded those visiting physicians (30.2 ± 66.9 versus 8.3 ± 13.0), the difference was not statistically significant.

All physicians and 16 healers announced their intention to continue using amygdalin in the future. Two healers indicated a switch to other natural compounds in the future. Amygdalin was applied in addition to conventional treatment in most cases (“always”: 21%, “mostly”: 66%) with no statistical differences between healers and physicians. Only three healers administered amygdalin to patients who did not receive conventional treatment (Table 3).

Amygdalin was primarily provided through pharmacies (*n* = 25; 66%), with a higher proportion among physicians (*n* = 16; 80%) compared to healers (*n* = 9; 50%; significant with *p* = 0.03). Sixteen participants (42%) indicated providing amygdalin directly from the manufacturer, with healers (*n* = 10; 56%) surpassing physicians (*n* = 6; 30%), although the difference was not statistically significant. Online provision was of minimal importance (*n* = 2; 5%) (Table 4).

Treatment for a particular cancer type was not indicated. The most commonly treated types were breast cancer (17%), prostate cancer (12%), and gastrointestinal tumors (12%).

Most study participants were convinced of amygdalin’s efficacy (all: 79%; healers: 78%; physicians: 80%). Accordingly, the majority of healers indicated amygdalin as the treatment of choice (61%). This was not the case among physicians, with only 25% expressing a preference for amygdalin, while 20% recommended alternative natural compounds. The difference between the groups was statistically significant (*p* = 0.01). Interestingly, nearly 30% of the participants reported that amygdalin therapy was started upon patient request. Additional comments (“Other”) were given by 21 participants. Of these, 7 participants noted laboratory test results as a prerequisite for amygdalin administration (e.g., evidence of circulating tumor cells). Twenty-one additional comments were provided. Five comments accentuated a patient-dependent amygdalin therapy (histology, staging, symptoms, physical condition of the patients). Seven comments emphasized the necessity of applying amygdalin in combination with further substances (e.g., vitamin C, resveratrol, mistletoe, dichloroacetic acid). Amygdalin was also administered when conventional treatment had been completed, between chemotherapy cycles, or when chemotherapy had not been tolerated (*n* = 2). The patient’s financial situation was mentioned as an exclusion/inclusion parameter (*n* = 3) (Table 5).

Nearly all providers (*n* = 34) administered amygdalin with the hope of delaying tumor progression. Symptom relief was indicated by 21 participants as an important reason for using amygdalin. Interestingly, 11 healers (61%) but only 5 physicians (25%) used amygdalin with the hope of a cure. The difference was statistically significant (*p* = 0.01). The greatest benefit of amygdalin was expected when administered in the early phase of the disease (*n* = 19; 50%). Twenty participants (52%) assumed maximum effects of amygdalin across all tumor stages (physicians, 70% > healers, 33%; *p* = 0.01). Few participants were convinced of amygdalin’s potency in the late tumor stage (*n* = 4) or in a palliative situation (*n* = 2). Additional comments were given in two cases (“Other”), pointing to the relevance of amygdalin in preventing recurrence (*n* = 1) or reducing chemotherapy (*n* = 1) (Table 6).

### 3.3. Information Sources

Initial information about amygdalin was primarily obtained from colleagues (*n* = 21; 55%) and congresses/congress reports (*n* = 18; 47%). In over 30% of cases (*n* = 12), initial information was provided by the patients themselves. Media, the internet, and scientific journals were less important sources compared to colleagues (*p* = 0.00).

Healers and physicians similarly continued to educate themselves on naturopathy, predominantly by means of colleagues (*n* = 32; 84%). Research through reference books was favored over that through scientific journals (*n* = 31 versus *n* = 20; *p* = 0.00). Internet research and congress visits were also noted as important educational tools. Physicians (*n* = 16; 80%) were significantly more inclined than healers (*n* = 7; 39%; *p* = 0.00) to use the internet for research (Table 7).

### 3.4. Patient Communication

Patient counseling was rated as “very good” by over 90% of amygdalin providers, with no difference between physicians and healers. Three providers reported that their patients already had “very good knowledge” of amygdalin and, hence, did not require further information. The majority of study participants (*n* = 29; 76%) believed that the patients would rank their advice as “very good”, and only a minority rated the advice from the patients’ perspective as “good” (*n* = 9; 24%). The answer options “neutral, “bad” or “don’t know, cannot assess” were not selected (Table 8).

### 3.5. Therapeutic Strategy

Amygdalin was administered i.v. by all participants (100%). Additionally, six healers (33%) and six physicians 30%) also applied it orally. Intravenous application was considered optimal by nearly all participants (*n* = 35; 92%). However, two healers and one physician were convinced that both oral uptake and infusion might be the most effective treatment strategy (Table 9).

A standardized treatment protocol did not exist. In most cases, amygdalin was given 1–2× per week (*n* = 10) or 5× per week (*n* = 10), and the healers (n = 7; 39%) preferred the 5×/week schedule compared to the physicians (*n* = 3; 15%; *p* = 0.05). Based on patient characteristics, three healers and two physicians additionally reported adjusting the therapy individually (Table 10, upper part).

Treatment predominately extended over several weeks and was carried out in regular intervals (*n* = 28) or patient-dependently (*n* = 9), with no significant differences between the physicians and healers. Four participants (two healers and two physicians) indicated (“Other”) that the frequency of amygdalin applications per week was reduced over time (e.g., from 5 applications/week to 1–2 applications per week or even to 1 application per month). Six participants always used the same amygdalin dosage, while the majority (*n* = 29) increased the dosage when required (Table 10 lower part).

Table 11 shows the initial amygdalin dosage and the final amygdalin dosage. Overall, mean initial values were 5.6 ± 2.7 g amygdalin per infusion, reaching a mean maximum of 14.3 ± 5.5 g amygdalin per infusion (no significant differences between physicians and healers). Most applicants started with 3 g amygdalin (16; 42%) and increased the dosage up to 9 g/infusion (*n* = 10; 26%). A final dosage of 15 g (*n* = 8; 21%) or 18 g (*n* = 7; 18%) was also frequently applied. Two healers increased the dosage to a maximum of 27 g. No significant difference between applied dosages was apparent between physicians and healers.

The most common reason for increasing the amygdalin dosage was “no or reduced therapeutic response” (*n* = 17; 45%). Dose escalation was also indicated when the patient’s condition had “worsened” (*n* = 8; 21%). Ten participants included further comments (“Other”). In this context, five participants increased the dosage according to the “therapeutic protocol” and five applicants enhanced the dosage based on tolerability (Table 12).

The applied therapeutic protocol was mainly based on recommendations given by colleagues (*n* = 27; 71%). Fifty percent of the participants also pointed to congress visits and congress reports as relevant information sources. A similar percentage of participants noted that treatment was carried out based on “own experience” (*n* = 18; 47%). Scientific literature was of minor importance (*n* = 4; 11%; *p* = 0.00). Three comments (“Other”) indicated that the amygdalin supplier was asked for advice, and two participants referred to “clinical studies” where they had obtained information about the dosing schedule of amygdalin. There was no significant difference between the response patterns of healers and physicians (Table 13).

### 3.6. Control of Toxic Effects and Therapeutic Response to Amygdalin

Seventeen physicians (85%) and 14 healers (78%) did not observe negative side effects following amygdalin infusion, regardless of the concentration used. Seven participants noted moderate side effects (five of these also used amygdalin orally). One physician observed slight fatigue, another physician reported a case of low blood pressure, and one physician reported dizziness and nausea. Four healers referred to (rare) cases of dizziness and nausea. Of these, one healer reported dizziness following the first infusion, while one healer observed dizziness following the last infusion after repeated application.

The majority of amygdalin prescribers did not control blood or serum cyanide levels in their patients (*n* = 34; 90%); only four participants (11%) did so. Twenty-four participants (63%) stated that measuring cyanide might not be necessary. “Not to confuse the patient” was a further reason why cyanide levels were not checked. Importantly, six participants added that metabolization to cyanide does not occur when amygdalin is given i.v., making cyanide control unnecessary. Other comments were given by four participants, with one healer arguing that “chemotherapy will also not be controlled”. Two healers “didn’t know that cyanide control might be important”. One physician asked “which laboratory will check cyanide”. Five prescribers did not answer this question (Table 14).

Laboratory data served to control treatment success (*n* = 31; 82%). However, no detailed information was given on this issue by eight prescribers. Ten prescribers pointed to “tumor markers” without further explanation. Seven physicians but only one healer monitored established specific (e.g., CEA) and non-specific (e.g., lactate dehydrogenase) parameters. One physician (additionally) and four healers (exclusively) assessed therapeutic response with the non-established EDIM test (Epitope Detection in Monocytes). Clinical examination and medical imaging were also frequently used to assess patient health. Patient-reported outcomes (*n* = 11; 29%) and physicians’ letters (*n* = 14; 37%) were less relevant. Statistical analysis did not reveal significant differences between healers and physicians (Table 15).

Amygdalin tablets were consumed orally at home. Asked whether patient compliance was controlled, 25 prescribers (66%) responded “no”. Eight prescribers (21%) did not respond to this question. Only five prescribers (13%) controlled their patients (Table 16). Limited communication between the amygdalin prescribers and the family doctor or medical specialist was reported. The majority of prescribers indicated that they rarely (*n* = 10, 26%) or never (*n* = 16, 42%) informed family doctors or medical specialists about the amygdalin treatment. The majority (*n* = 27; 71%) also did not provide information to the family physicians or medical specialists in regard to treatment efficacy (Table 16).

### 3.7. Amygdalin Efficacy

Amygdalin’s potential was rated “very high” by nine healers (*n* = 7, 39%) > physicians (*n* = 2, 10%; *p* = 0.02). Towards the other end of the valuation, healers and physicians together assessed amygdalin’s potential as only moderate (*n* = 7, 18%). Most rated amygdalin’s potential as “medium” (*n* = 17, 45%). Notably, four providers (11%) could not assess amygdalin’s potential, and two providers did not respond. There was no significant difference between the responses of healers and physicians (Table 17).

Several factors were associated with the therapeutic success of amygdalin, with a correct indication of the disease (n = 25; 66%), correct amygdalin dosing (n = 22; 58%), and additional dietary change (*n* = 26; 68%) being the most mentioned parameters. In an additional comment (“Other”), several providers emphasized that amygdalin should be embedded in a multimodal concept (n = 17; 45%), including the use of further natural substances (e.g., curcumin, artesunate, vitamins), detoxification, immunostimulation, and/or psychotherapy (Table 17).

### 3.8. Scientific Knowledge and Expectations

The quality of amygdalin-related information sources was assessed on a scale of 0 (=very poor) to 10 (=very good). Exchanges with colleagues and naturopathic congresses were rated highest by both healers and physicians. Healers (but not physicians) also felt well informed by journals (the term “journals” was not further specified). Medical congresses were deemed less informative compared to naturopathic congresses (significant with healers; *p* = 0.01). Information on amygdalin provided by standard medical books was rated poor by both healers and physicians (significant difference in all categories, i.e., colleagues, journals, medical and naturopathic congresses) (Figure 1).

When asked about future expectations, amygdalin providers criticized the limited attention and acceptance of naturopathic treatment options in conventional medicine. Most importantly, clinical studies were requested (*n* = 34; 90%), and nearly all amygdalin providers (*n* = 35; 92%) requested open-mindedness from medical colleagues toward naturopathic treatment concepts. Thematic debating by a scientific reference center (*n* = 24; 63%) and consideration of natural compounds in medical guidelines (*n* = 23; 61%) were also highly desired. Significantly more physicians than healers (*n* = 18; 90% versus *n* = 11; 61%; *p* = 0.02) asked for greater incorporation of naturopathy in medical congresses (*n* = 29; 76%). In-depth consideration of naturopathy in standard medical books (*n* = 16; 42%), however, was deemed less relevant (Table 18).

The majority of providers expressed a need for continuous information on new study results (*n* = 28; 74%) and a reputable platform or forum (*n* = 28; 74%) that allows scientific exchange on naturopathy. This forum should also offer the opportunity to present novel ideas and scientific concepts (*n* =13; 34%). Future support was also desired concerning the analysis of active components (*n* = 9; 24%), bioavailability (*n* = 11; 29%), and tolerability (*n* = 11; 29%) of natural drugs (Table 19).

Nonetheless, only a few providers (*n* = 10; 26%) indicated that they evaluated patient data themselves; the majority did not (*n* = 28; 74%). Similarly, only six providers (16%) noted that they had been involved in a former study, whereas 32 (84%) had not. Twenty-two physicians and healers (58%) expressed a desire to participate in a future study. However, 12 participants refused future participation and four participants (11%) did not respond. When asked whether the participants would be willing to anonymously provide patient data for scientific research, only 13 participants agreed (34%), 12 participants would not, and 13 participants did not respond. In the healer group, only five (28%) declared willingness to provide data for scientific analysis, whereas 13 healers refused or did not respond to this question (Table 20).

## 4. Discussion

Nearly 20 cancer patients per practitioner were treated quarterly with amygdalin infusions. Extrapolating this figure to all 76 practitioners identified in our study, more than 1500 cancer patients receive amygdalin infusions quarterly. Given that new patients are treated each quarter, more than 6000 patients receive amygdalin infusions per year. Accounting for undetected prescribers, the actual number of cancer patients receiving intravenous amygdalin treatment could be substantially higher. A considerable number of patients could also be ingesting amygdalin tablets or apricot kernels without medical supervision. Considering the widespread use of amygdalin, little pertinent publication is available. Treatment through physicians and healers in Germany was investigated for the first time in the present study.

A cross-sectional study consisting of 166 Malaysian cancer patients using CAM indicated that oral amygdalin was “the most common dietary supplement” [15]. A Polish survey conducted on internet forums and social networking sites found that among 177 CAM users, 50 cancer patients consumed amygdalin (28.3%) [3]. The percentage of amygdalin users among inpatients and outpatients undergoing oncological treatment was calculated to be 5.3% [16], while 10.8% of hospitalized Polish cancer patients who were aware of CAM declared amygdalin use [17]. Information regarding amygdalin use in Germany is sparse. A survey on outpatient cancer patients has reported amygdalin use at 5.5% [18].

Despite the lack of evidence supporting amygdalin efficacy in treating cancer and warnings regarding the risk of poisoning, its use by patients appears to be high [19,20]. The rationale behind the preference for amygdalin over other CAM options can only be speculated upon. While nearly all the study participants rated their communication with patients as excellent and believed that their patients also felt well-informed about amygdalin, it is possible that patients are influenced by practitioners convinced of amygdalin’s therapeutic potential. Facing advanced cancer, desperate and willing to try anything that may help, patients are susceptible to any promise of a cure [21]. Goals and expectations must nevertheless be explained realistically, enabling patients to make informed, autonomous decisions about whether to use amygdalin or pursue alternative options.

The main reasons specified by the practitioners for administering amygdalin were symptom relief and delaying disease progression. Still, a number of them, particularly healers, indicated they administered amygdalin with the intent to cure. To date, there is no evidence to support the claim that amygdalin can cure cancer. The current survey does not address whether the belief in a cure was communicated to the patient. Without question, it is imperative to avoid making unrealistic promises to patients that could foster false hopes and unrealistic expectations. Striking a balance between achievability and unattainability is essential but also challenging, given that most available information on amygdalin is derived from antiquated studies. Undoubtedly, further research is warranted to provide a realistic assessment of amygdalin’s potential.

Therapeutic recommendations based on complementary and alternative medicine (CAM) may be influenced by economic interests. Indeed, three physicians mentioned the financial situation of patients as a limiting factor for initiating amygdalin infusions. Based on an internet search, a 3 g amygdalin vial for i.v. application costs less than 10 € [22,23], but in Germany, one pharmacy offers a 3 g vial for 30 € [24]. Consequently, costs for one week’s treatment range from 150 € (3 g infusion/day) to 300 € (6 g infusion/day). Total treatment costs, including application and supervision, may, however, be higher.

Amygdalin was not always considered to be the treatment of choice. Some practitioners initiated amygdalin therapy solely due to patient requests. Other studies have identified insufficient knowledge of CAM and unsatisfactory dialogue with clinicians about CAM use as relevant factors contributing to patient demand [25,26]. Not requiring further consultation with their providers raises the question of how these patients obtain information about amygdalin. Clinically sound data with regard to amygdalin are scarce, with only two low-standard amygdalin trials having been conducted more than 40 years ago. These trials cannot serve as guidelines to evaluate amygdalin in the 21st century [12]. Given the public interest in amygdalin and the uncertainties surrounding its efficacy, high-standard patient studies meeting randomized controlled trial criteria should be conducted. Presently, tumor patients tend to seek advice from family, friends, or other tumor patients, or they search the internet for information [27,28,29].

Practitioners encounter similar challenges in regard to information about amygdalin. Initially, information is primarily obtained from colleagues and congresses, with medical congresses being perceived as less informative compared to naturopathic congresses. More than 30% of amygdalin providers reported that they obtained initial information about amygdalin from their patients. Therefore, amygdalin treatment concepts may be based on unverified anecdotal reports, mistakenly perceived as scientifically sound. The lack of evidence-based studies makes it difficult to distinguish between “serious” and “unserious” information. Despite growing interest, CAM remains under-represented in national and international cancer congresses. The healers and physicians in the present study advocate for open-mindedness among medical colleagues toward naturopathic treatment concepts. Indeed, efforts have been made in recent years to integrate CAM into cancer treatment services [30,31]. Still, a reputable forum for scientific exchange between medical and non-medical practitioners or between CAM users and non-CAM users has not been established.

In the present study, intravenous rather than orally administered amygdalin was considered more effective. However, it is unclear where this preference originated. Only one peer-reviewed manuscript has been published dealing with i.v. amygdalin to treat cancer patients [32]. Therefore, it must be assumed that all information concerning i.v. amygdalin dosage has been taken from this report. Still, even the concentration used in this trial was not established in pilot tests but was rather determined from a “published compendium” of “some leading” practitioners treating with amygdalin [32]. Respective information provided by Moertel et al. is also confusing. Although vials containing 3 g amygdalin for i.v. injections were ordered, and the treatment regimen pointed to an amygdalin concentration of 4.5 g/m^2^ body surface area daily for three weeks. Presumably, two vials (6 g) of amygdalin were given per day. Contreras, a leading amygdalin pioneer, administered variable dosages ranging from 1 to 10 g amygdalin daily [12,33] but finally remarked that 6 g amygdalin i.v. daily was the most convenient method. There are also early reports on i.v. dosages of <1 g, 3, or 9 g amygdalin per day, applied by other amygdalin providers [12,34].

In the present investigation, the initial dose of 3 g amygdalin was most frequently applied with an increase to a maximum of 9 g/day in the majority of cases. This is within the range of the concentrations applied by the amygdalin pioneers. However, over 60% of providers treated their patients with >9 g/day amygdalin, a practice not supported by early reports. Dosage was not tied to the body surface area as practiced by Moertel et al. [12]. Rather, amygdalin was applied in 3 g units, i.e., 3 g or a multiple of 3 g. Nearly half of the amygdalin appliers based their dosage decisions on their “own experience”. Elevating the amygdalin dose was particularly common in cases of limited response or worsened condition. This approach carries significant risk, as a loss of response to treatment may indicate either a lack of anti-tumor effect or the development of resistance. Elevating amygdalin dosage under these circumstances could further exacerbate the patient’s precarious condition.

Amygdalin was administered at regular intervals in most cases, which aligns with the protocol introduced by Moertel et al., which involves intravenous injections over three weeks (on consecutive days or weekdays only). Moertel et al. additionally pointed to the published recommendations of Ernest Krebs, Jr., who developed the injectable form of amygdalin and stated that intravenous treatment should be administered over periods ranging from 10 to 31 days [32].

In current clinical practice, monitoring serum tumor markers is common for assessing therapeutic response. According to the surveyed practitioners, laboratory data was considered the most relevant factor for evaluating treatment success. Still, the term “laboratory data” was often unspecified, leaving room for non-established tests to be used. Five amygdalin providers assessed therapeutic response by the EDIM test (Epitope Detection in Monocytes). This test quantifies the expression level of the enzyme transketolase-like 1 (TKTL1) in monocytes and is popular among alternative practitioners but plays no role in conventional medicine. A recent study on patients undergoing curative radical prostatectomy demonstrated no correlation between serum TKTL1 and serum PSA, Gleason score, tumor stage, or further clinical and pathologic parameters [35]. Hence, TKTL1 should not be considered a specific biomarker, as misinterpretation could lead to erroneous treatment decisions. It is essential to monitor patient response rigorously, using well-established guideline-based tests, especially given amygdalin’s status as a non-standard therapy with unknown efficacy. Encouragingly, 65% of the surveyed practitioners monitored therapeutic success through clinical examination and/or medical imaging, a practice worthy of commendation.

Good communication between amygdalin providers and other healthcare providers is absent. Many physicians have limited knowledge about CAM, hampering dialog between naturopaths and physicians. This disconnect extends to the perceived irrelevance of physicians' letters in guiding therapeutic decisions, and open-mindedness is called for on both sides. Acknowledging that differing perspectives on complementary treatment methods may exist between CAM providers and healthcare providers who do not utilize CAM could optimize patient outcomes.

Questions concerning amygdalin’s value as an anti-cancer drug have been raised in regard to the risk of HCN poisoning. In this survey, severe adverse events were not reported. A few cases of dizziness and nausea were observed, but these were independent of amygdalin concentration or application time. This aligns with findings from the Moertel study, demonstrating no clinical or laboratory evidence of toxicity when amygdalin was given intravenously at 6 g/day [11]. Another study evaluating cyanide levels after chronic oral amygdalin intake and subsequent intravenous amygdalin administration also found no signs of HCN intoxication, such as nausea, dizziness, or loss of consciousness [36]. Similarly, no serious adverse events have been recorded when healthy subjects have repeatedly been given intravenous amygdalin infusions [37].

It cannot be excluded that the (moderate) negative effects recorded in the present investigation were caused by conventional therapy. However, this seems unlikely since patients were questioned about negative side effects observed during the actual amygdalin infusion. One participant noted nausea that disappeared after 1 h. Another reported fatigue that disappeared after 4 h, and one patient complained of a mild headache that disappeared when the infusion was slowed. While amygdalin infusion appears to be safe for treating cancer patients, careful monitoring during and after treatment is important. Fortunately, amygdalin infusions are only administered in the healer’s or physician’s practice, ensuring proper oversight.

A significant concern arises with oral amygdalin consumption at home, especially considering the limited follow-up reported by only 13% of the surveyed practitioners. A proper maintenance dosage typically involves ingesting 1–3 tablets containing 0.5–1.5 mg of amygdalin per day [32]. In a small pilot study, no clinical or laboratory evidence of toxic reaction was observed in patients taking 0.5 g amygdalin orally three times daily [11]. However, headaches, dizziness, nausea, and vomiting have been observed in a few patients following the consumption of 3 tablets (1.5 g amygdalin) per day [32]. Another study reported no HCN-related side effects with daily doses of 0.5–2 g for 2–43 weeks [38]. Despite these findings, several cases of severe toxicity and even death after oral amygdalin consumption have been documented, which, however, were related to substantial overdosing and/or additional chewing of a large quantity of apricot kernels [12]. The risk of cyanide poisoning from uncontrolled amygdalin consumption at home should, therefore, not be underestimated. Particularly the argument that “the more, the better” may drive patients to increase their dosage arbitrarily without informing their healthcare provider.

The reason for the lack of compliance control can only be speculated upon. It is not clear whether amygdalin providers are aware of the risk associated with amygdalin. Possibly, their daily workload does not allow for a careful follow-up. A low-interest level might also be a reason why patients were not controlled at home. In fact, this survey documents a lack of communication between the amygdalin providers and other healthcare providers, along with a limited interest in patient-reported outcomes and physicians’ letters. Due to the risk of overdosing, frequent monitoring in the form of telephone calls or messaging apps [39,40] after patients leave the practice should be mandatory.

Ames et al. and Moertel et al. [11,41] reported that amygdalin is rapidly cleared from the blood and excreted largely unchanged in the urine when administered i.v. Nearly all practitioners in the present survey deemed it unnecessary to determine blood HCN levels since amygdalin is not metabolized to HCN when given i.v. In vivo studies conducted on rats and dogs point to an amygdalin excretion rate of approximately 80% [42], which has been verified in healthy men [37]. The lack of amygdalin metabolization to HCN might explain why no HCN-related side effects have been observed in cancer patients following i.v. amygdalin treatment. However, this opens the question about the anti-tumor mechanism of amygdalin, assumed to be due to HCN specifically accumulating in cancer cells. All surveyed practitioners were convinced that i.v. amygdalin application was the most effective treatment method. However, under the assumption that HCN is relevant to destroying cancer cells, oral consumption of amygdalin (producing high amounts of HCN) might be superior to i.v. treatment. Only a minority of amygdalin providers monitored their patients consuming tablets orally at home, making it difficult to draw definitive conclusions. Studies by Mani et al. detected an increase in the HCN blood level following amygdalin injection [36], indicating that at least some degree of metabolization may occur independently of gut flora activity. However, this remains speculative. The discrepancy between the assumed mode of action of amygdalin and the lack of HCN synthesis after i.v. application calls for further research. In vitro studies have shown that amygdalin induces apoptosis, suppresses proliferation, and inhibits the metastatic spread of cancer cells. Recent experiments by Zhang et al. on rats have proposed that amygdalin is deglycosylated to prunasin, its active metabolite [43]. Other approaches point to the protective effects of amygdalin on epithelial–mesenchymal transformation in mice [44,45]. There is also evidence that amygdalin inhibits prostate [46], lung [47], and liver [48] cancer growth in vivo. These reports open new avenues for understanding amygdalin’s mechanism of action and highlight the urgency of conducting animal and clinical studies to validate these findings.

The physicians and healers participating in this survey recognized the need for more effort leading to a realistic assessment of amygdalin’s value. Enrolling in clinical trials on CAM was one of the most requested actions. It is, therefore, disappointing that the interest in actively participating in future studies was moderate, with only one-third of the surveyed individuals agreeing to provide data for scientific analysis. Whether these non-interested providers were actually overloaded with work, whether they did not want to make their (anonymously submitted) data public, or whether they feared a negative impact on the future use of amygdalin remains unclear. Assessing the effectiveness of amygdalin for cancer care remains limited as long as relevant study data are unavailable. Hence, amygdalin providers should become more engaged in scientific research to open fruitful discussions based on evidence rather than speculation. While it may be unrealistic to expect a randomized controlled trial, single case reports may also provide valuable information.

Limitations of this study should be considered. Most of the patients taking amygdalin also received conventional therapy. Therefore, effects thought to be due to amygdalin (e.g., Table 17) could have been due to conventional treatment. Some responses (e.g., Table 8) particularly reflect the personal opinion of the practitioner and might be unduly positive. A new study is therefore aimed at attaining information from the patient’s point of view. It will include not only the patient’s assessment of amygdalin’s anti-tumor potential, assessment of counseling, and overall satisfaction but also long-term observation and patient-reported outcome.

## 5. Conclusions

This survey documents that, supplementary to conventional treatment, a considerable number of tumor patients are being treated with amygdalin. The application of amygdalin is based on protocols developed by pioneers but often modified by practitioners. Severe HCN-induced toxicity was not observed, presumably because amygdalin i.v. is excreted nearly unmetabolized. Therefore, amygdalin infusion appears to be safe, while oral ingestion should be monitored. Dialogue between amygdalin users and their physicians is unsatisfactory and requires improvement. Despite amygdalin infusions being considered the most effective, the lack of HCN synthesis associated with this application method questions amygdalin’s presumed mode of action. Amygdalin providers are encouraged to become more engaged in scientific research, which is necessary to realistically evaluate amygdalin’s anti-cancer potential and determine its risk–benefit ratio.

## Figures and Tables

**Figure 1 nutrients-16-02068-f001:**
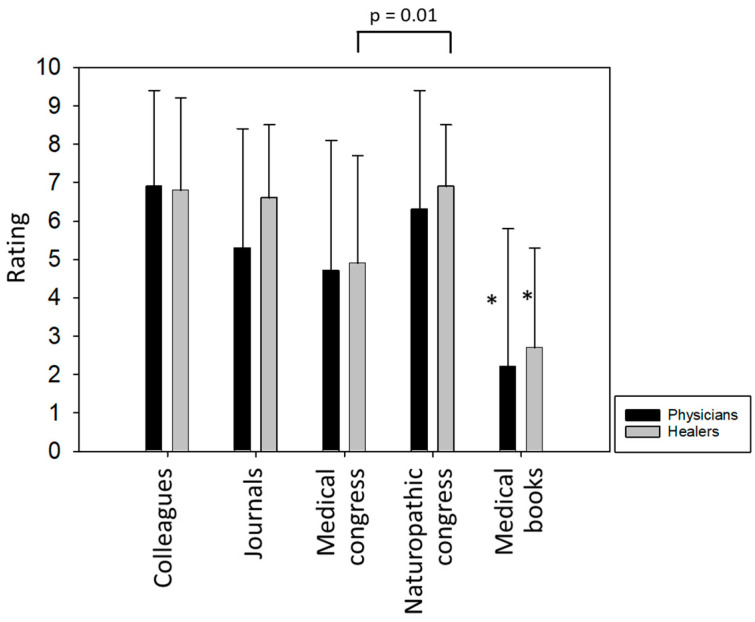
Rating of amygdalin-related information sources. * significant difference in all categories.

**Table 1 nutrients-16-02068-t001:** Gender and age.

	Sex [(n (%)]		Age (Years)		
Category	Males	Females	Mean ± SD	Min	Max
All	26 (68)	12 (32)	58.3 ± 8.1	37	81
Physicians	17 (85)	3 (15)	59.0 ± 7.5	40	71
Healers	9 (50)	9 (50)	57.4 ± 8.7	37	81

**Table 2 nutrients-16-02068-t002:** Additional knowledge.

	Education [(n (%)] ^a^								
Category	Acupuncture ^b^	Naturopathy ^b^	Nutritional Medicine	Homeopathy ^b^	Physical Therapy	Manual Medicine	Psychotherapy	MTT ^c^	Palliative Medicine
All	8 (21)	15 (40)	9 (24)	9 (24)	4 (11)	8 (21)	3 (8)	8 (21)	4(11)
Physicians	7 (35)	11 (55)	5 (25)	8 (40)	2 (10)	6 (30)	0 (0)	4 (20)	0 (0)
Healers	1 (6)	4 (22)	4 (22)	1 (6)	2 (11)	2 (11)	3 (17)	4 (22)	4(22)

^a^ Multiple answers allowed. ^b^ Significant (physicians versus healers): Acupuncture: *p* = 0.01; Naturopathy: *p* = 0.03; Homeopathy: *p* = 0.01. ^c^ MTT: Medical Tumor Therapy.

**Table 3 nutrients-16-02068-t003:** Duration of amygdalin use and number of patients treated.

	Amygdalin Use ^a^	Patients/Quarter ^a^	Continue Amygdalin ^b^		Conventional Treatment ^b^			
Category	(Years)	(Number)	Yes	No	Always	Mostly	Not	NR
All	8.8 ± 5.5	18.6 ± 48.2	36 (95)	2 (5)	8 (21)	25 (66)	3 (8)	2 (5)
Physicians	8.8 ± 6.2	8.3 ± 13.0	20 (100)	0 (0)	6 (30)	12 (60)	0 (0)	2 (10)
Healers	8.8 ± 4.7	30.2 ± 66.9	16 (89)	2 (11)	2 (11)	13 (72)	3 (17)	0 (0)

^a^ Mean ± SD. ^b^ Values given as number (first value) and percentage (in brackets). Percentage values are rounded without decimal places. NR: No response.

**Table 4 nutrients-16-02068-t004:** Ordering of amygdalin.

Category	Pharmacy ^a,b^	Manufacturer	Internet
All	25 (66)	16 (42)	2 (5)
Physicians	16 (80)	6 (30)	2 (10)
Healers	9 (50) ^c^	10 (56)	0 (0)

^a^ All values given as number (first value) and percentage (in brackets). Percentage values are rounded without decimal places. ^b^ Multiple answers allowed. ^c^ Significant: *p* = 0.03 (healers versus physicians).

**Table 5 nutrients-16-02068-t005:** Reason for amygdalin use.

Category	Convinced ^a,b^	Patient Wish	NR	First Choice	Not First Choice	Other
All	30 (79)	11 (29)	3 (8)	16 (42)	5 (13)	21 (55)
Physicians	16 (80)	6 (30)	1 (5)	5 (25)	4 (20)	9 (45)
Healers	14 (78)	5 (28)	2 (11)	11 (61) ^c^	1 (6)	12 (67)
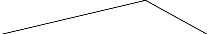
**Category**	**Lab Test ^a,b^** **Positive**	**Patient** **Dependent**	**Additional** **Compounds**	**Financial** **Situation**	**Therapeutic Break**
All	7 (18)	5 (13)	7 (18)	3 (8)	2 (5)
Physicians	3 (15)	2 (10)	3 (15)	3 (15)	1 (5)
Healers	4 (22)	3 (17)	4 (22)	0 (0)	1 (6)

^a^ All values given as number (first value) and percentage (in brackets). ^b^ Multiple answers allowed. ^c^ Significant: *p* = 0.01. NR: No response.

**Table 6 nutrients-16-02068-t006:** Expectations of amygdalin’s potential.

Category	Symptom Relief ^a,b^	Progression Delay	Cure	NR	Early Stage ^a,b^	Advanced Stage	All Stages	Palliative	Other
All	21 (55)	34 (90)	16 (42)	1 (3)	19 (50)	4 (11)	20 (53)	2 (5)	2 (5)
Physicians	13 (65)	18 (90)	5 (25)	1 (5)	9 (45)	1 (5)	14 (70)	2 (10)	1 (5)
Healers	8 (44)	16 (89)	11 (61) ^c^	0 (0)	10 (56)	3 (17)	6 (33) ^c^	0 (0)	1 (6)

^a^ All values given as number (first value) and percentage (in brackets). Percentage values are rounded without decimal places. ^b^ Multiple answers allowed. ^c^ Significant: *p* = 0.01 (healers versus physicians). NR: No response.

**Table 7 nutrients-16-02068-t007:** Initial and continuous information sources on amygdalin.

Initial	Colleagues ^a,b^	Internet, Media	Scientific Literature	Congress (Reports)	Patients	Other
All	21 (55) ^c^	10 (26)	8 (21)	18 (47)	12 (32)	1 (3)
Physicians	11 (55)	7 (35)	3 (15)	7 (35)	8 (40)	0 (0)
Healers	10 (56)	3 (17)	5 (28)	11 (61)	4 (22)	1 (6)
**Continued**	**Colleagues ^a,b^**	**Internet**	**Reference Books**	**Scientific Journals**	**Congress**	**Other**
All	32 (84)	23 (60)	31 (82) ^d^	20 (53)	26 (68)	2 (5)
Physicians	18 (90)	16 (80) ^e^	16 (80)	12 (60)	14 (70)	1 (5)
Healers	14 (78)	7 (39)	15 (83)	8 (44)	12 (67)	1 (6)

^a^ All values given as number (first value) and percentage (in brackets). Percentage values are rounded without decimal places. ^b^ Multiple answers allowed. ^c^ Significant: *p* = 0.00 (colleagues all versus internet, literature all). ^d^ Significant: *p* = 0.00 (literature general all versus literature scientific all). ^e^ Significant: *p* = 0.00 (internet physicians versus internet healers).

**Table 8 nutrients-16-02068-t008:** Assessment of information quality and assessment of patient satisfaction.

Category	Information by User ^a^		Patients‘ Satisfaction ^a^	
	Very Good	Not Necessary ^b^	Very Good	Good
All	35 (92)	3 (8)	29 (76)	9 (24)
Physicians	18 (90)	2 (10)	15 (75)	5 (25)
Healers	17 (94)	1 (6)	14 (78)	4 (22)

^a^ All values given as number (first value) and percentage (in brackets). Percentage values are rounded without decimal places. ^b^ Indicated reason: Patients already had a very good knowledge.

**Table 9 nutrients-16-02068-t009:** Kind of amygdalin application and optimum treatment conditions.

Category	Application ^a^		Optimum Treatment	
	Oral	i.v.	Oral	i.v.
All	12 (32)	38 (100)	3 (8)	35 (92)
Physicians	6 (30)	20 (100)	1 (5)	19 (95)
Healers	6 (33)	18 (100)	2 (11)	16 (89)

^a^ All values given as number (first value) and percentage (in brackets). Percentage values are rounded without decimal places.

**Table 10 nutrients-16-02068-t010:** Amygdalin treatment schedule.

Category	1×/Week ^a^	1–2×/Week	2–3×/Week	3–5×/Week	5×/Week	Individually	NR
All	2 (5)	10 (26)	6 (16)	4 (11)	10 (26)	5 (13)	1 (3)
Physicians	2 (10)	5 (25)	4 (20)	3 (15)	3 (15)	2 (10)	1 (5)
Healers	0 (0)	5 (28)	2 (11)	1 (6)	7 (39) ^b^	3 (17)	0 (0)
**Category**	**Only One Cycle ^a,b^**	**Refular Intervals**	**Patient Dependent**	**Other**	**Dosage**		
					**Constant**	**Increasing**	**NR**
All	3 (8)	28 (74)	9 (24)	4 (11)	6 (16)	29 (76)	3 (8)
Physicians	2 (10)	15 (75)	3 (15)	2 (10)	3 (15)	15 (75)	2 (10)
Healers	1 (6)	13 (72)	6 (33)	2 (11)	3 (17)	14 (78)	1 (6)

^a^ All values given as number (first value) and percentage (in brackets, rounded without decimal places). ^b^ Significant: *p* = 0.05 (healers versus physicians).

**Table 11 nutrients-16-02068-t011:** Initial and final dosages of applied amygdalin.

Start	3 g ^a^	6 g		7.5 g	9 g		12 g	NR
All	16 (42)	8 (21)		1 (3)	9 (24)		1 (3)	3 (8)
Physicians	10 (50)	4 (20)		0 (0)	3 (15)		1 (5)	2 (10)
Healers	6 (33)	4 (22)		1 (6)	6 (33)		0 (0)	1 (6)
**End**	**3 g ^a^**	**9 g**	**12 g**	**15 g**	**18 g**	**24 g**	**27 g**	**NR**
All	1 (3)	10 (26)	5 (13)	8 (21)	7 (18)	2 (5)	2 (5)	3 (8)
Physicians	1 (5)	5 (25)	2 (10)	4 (20)	4 (20)	2 (10)	0 (0)	2 (10)
Healers	0 (0)	5 (28)	3 (17)	4 (22)	3 (17)	0 (0)	2 (11)	1 (6)

^a^ All values given as number (first value) and percentage (in brackets, rounded without decimal places). NR: No response.

**Table 12 nutrients-16-02068-t012:** Reasons for dose escalation.

Category	Limited Response ^a,b^	Worsened Condition	Other		NR
			Protocol	Tolerability	
All	17 (45)	8 (21)	5 (13)	5 (13)	5 (13)
Physicians	8 (40)	5 (25)	3 (15)	2 (10)	3 (15)
Healers	9 (50)	3 (17)	2 (11)	3 (17)	2 (11)

^a^ All values given as number (first value) and percentage (in brackets). Percentage values are rounded without decimal places. ^b^ Multiple answers allowed. NR: No response.

**Table 13 nutrients-16-02068-t013:** Therapy-relevant information sources.

Category	Colleagues ^a,b^	Internet, Media	Scientific Literature	Congress (Reports)	Patients	Own Experience	Other	
							Studies	Supplier
All	27 (71) ^c^	6 (16)	4 (11)	19 (50)	3 (8)	18 (47) ^c^	2 (5)	3 (8)
Physicians	15 (75)	5 (25)	1 (5)	10 (50)	3 (15)	8 (40)	0 (0)	1 (5)
Healers	12 (67)	1 (6)	3 (17)	9 (50)	0 (0)	10 (56)	2 (11)	2 (11)

^a^ All values given as number (first value) and percentage (in brackets). Percentage values are rounded without decimal places. ^b^ Multiple answers allowed. ^c^ Significant: *p* = 0.00 (all colleagues versus all scientific literature; all own experience versus all scientific literature).

**Table 14 nutrients-16-02068-t014:** Information regarding cyanide.

Category	Cyanide Control ^a^		Reason for No Control ^a,b^						
	Yes	No	Not Necessary	Patient Confusion	Cumbersome	Too Expensive	No Cyanide Release	Other	NR
All	4 (11)	34 (90)	24 (63)	7 (18)	1 (3)	3 (8)	6 (16)	4 (11)	5 (13)
Physicians	1 (5)	19 (95)	14 (70)	3 (15)	1 (5)	3 (15)	2 (10)	1 (5)	2 (10)
Healers	3 (17)	15 (83)	10 (56)	4 (22)	0 (0)	0 (0)	4 (22)	3 (17)	3 (17)

^a^ All values given as number (first value) and percentage (in brackets, rounded without decimal places). ^b^ Multiple answers allowed. NR: No response.

**Table 15 nutrients-16-02068-t015:** Control of therapeutic success.

Category	Clinical Examination ^a,b^	Imaging	Physician’s Letter	Patient Report	Laboratory Data	Lab Data—Examples ^c^			
						No	TM	ETM	NETM
All	25 (66)	25 (66)	14 (37)	11 (29)	31 (82)	8	10	8	6
Physicians	11 (55)	12 (60)	5 (25)	7 (35)	17 (85)	4	5	7	2
Healers	14 (78)	13 (72)	3 (17)	4 (22)	14 (78)	4	5	1	4

^a^ All values given as number (first value) and percentage (in brackets, rounded without decimal places). ^b^ Multiple answers allowed. ^c^ Values given as numbers. TM: Tumor marker; ETM: Established tumor marker; NETM: Not established tumor marker; No: no examples given.

**Table 16 nutrients-16-02068-t016:** Compliance control and information exchange.

Category	Compliance Control ^a^			Info on Amygdalin Use ^a^					Info on Amygdalin Response ^a^		
	No	Yes	NR	Always	Mostly	Rarely	No	NR	Yes	No	NR
All	25 (66)	5 (13)	8 (21)	3 (8)	7 (18)	10 (26)	16 (42)	2 (5)	8 (21)	27 (71)	3 (8)
Physicians	14 (70)	3 (15)	3 (15)	2 (10)	2 (10)	4 (20)	10 (50)	2 (10)	4 (20)	14 (70)	2 (10)
Healers	11 (61)	2 (11)	5 (28)	1 (6)	5 (28)	6 (33)	6 (33)	0 (0)	4 (22)	13 (72)	1 (6)

^a^ All values given as number (first value) and percentage (in brackets). Percentage values are rounded without decimal places. NR: No response.

**Table 17 nutrients-16-02068-t017:** Assessment of amygdalin potential and factors relevant to treatment.

Category	Potential of Amygdalin ^a,b^					Additional Factors ^a,b^				
	Very High	Medium	Moderate	Unclear	NR	Indication	Dosing	Nutrition	Lifestyle	Multimodal
All	10 (26)	17 (45)	7 (18)	4 (11)	2 (5)	25 (66)	22 (58)	26 (68)	17 (45)	17 (45)
Physicians	3 (15)	9 (45)	5 (25)	3 (15)	2 (10)	13 (65)	13 (65)	11 (55)	6 (30)	9 (45)
Healers	7 (39) ^c^	8 (44)	2 (11)	1 (6)	0 (0)	12 (67)	9 (50)	15 (83)	11 (61)	8 (44)

^a^ All values given as number (first value) and percentage (in brackets). Percentage values are rounded without decimal places. ^b^ Multiple answers allowed. ^c^
*p* = 0.02 (healers versus physicians). NR: No response.

**Table 18 nutrients-16-02068-t018:** Expectations of amygdalin providers in the medical community.

Category	Clinical Studies ^a,b^	Medical Congresses	Medical Books	Medical Guidelines	Reference Center	Open Mindedness
All	34 (90)	29 (76)	16 (42)	23 (61)	24 (63)	35 (92)
Physicians	16 (80)	18 (90)	9 (45)	12 (60)	12 (60)	17 (85)
Healers	18 (100)	11 (61) ^c^	7 (39)	11 (61)	12 (67)	18 (100)

^a^ All values given as number (first value) and percentage (in brackets). Percentage values are rounded without decimal places. ^b^ Multiple answers allowed. ^c^
*p* = 0.02 (healers versus physicians).

**Table 19 nutrients-16-02068-t019:** Future support desired by amygdalin prescribers.

Category	Active Component ^a,b^	Bioavailability	Tolerability	Study Results	Scientific Exchange	Presenting Ideas
All	9 (24)	11 (29)	11 (29)	28 (74)	28 (74)	13 (34)
Physicians	4 (20)	5 (25)	7 (35)	15 (75)	13 (65)	5 (25)
Healers	5 (28)	6 (33)	4 (22)	13 (72)	15 (83)	8 (44)

^a^ All values given as number (first value) and percentage (in brackets, rounded without decimal places). ^b^ Multiple answers allowed.

**Table 20 nutrients-16-02068-t020:** Willingness to participate in clinical studies.

Category	Evaluating Results ^a^		Study Participation ^a^		Future Participation ^a^			Provide Data ^a^		
	Yes	No	Yes	No	Yes	No	NR	Yes	No	NR
All	10 (26)	28 (74)	6 (16)	32 (84)	22 (58)	12 (32)	4 (11)	13 (34)	12 (32)	13 (34)
Physicians	4 (20)	16 (80)	3 (15)	17 (85)	13 (65)	5 (25)	2 (10)	8 (40)	7 (35)	5 (25)
Healers	6 (33)	12 (67)	3 (17)	15 (83)	9 (50)	7 (39)	2 (11)	5 (28)	5 (28)	8 (44)

^a^ All values given as number (first value) and percentage (in brackets). Percentage values are rounded without decimal places. NR: No response.

## Data Availability

Data are contained within the article.

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
