# Peer review of "Survey of Physicians and Healers Using Amygdalin to Treat Cancer Patients"

_nutrients, 2024, doi:10.3390/nu16132068_

Round 1
Reviewer 1 Report
Comments and Suggestions for Authors
Dear Authots,
Please add in the discussion area the following 3 paragraphs :
-Previous experience with the experimental use of amygdalin in laboratory animals.
- Limitations of your study.
- Future perspectives and studies from your team.
Comments on the Quality of English LanguageMinor editing of English language required.
Author Response
Answers to the comments of referee 1
Comment 1:
Please add in the discussion area the following 3 paragraphs :
Previous experience with the experimental use of amygdalin in laboratory animals.
Our answer: We have now added to the discussion (lines 603-606): “Other approaches point to protective effects of amygdalin on epithelial-mesenchymal transformation in mice [44,45]. There is also evidence that amygdalin inhibits prostate [46], lung [47], and liver [48] cancer growth in vivo. These reports open new avenues …”.
Limitations of your study.
We have now added to the discussion (lines 622-626): “Limitations of this study should be considered. Most of the patients taking amygdalin also received conventional therapy. Therefore, effects thought to be due to amygdalin (e.g. table 17) could have been due to conventional treatment. Some responses (e.g. table 8) particularly reflect the personal opinion of the practitioner and might be unduly positive. ”.
Future perspectives and studies from your team.
We have now added to the discussion (lines 626-629): “A new study is therefore aimed at attaining information from the patient’s point of view. It will include not only the patient’s assessment of amygdalin’s anti-tumor potential, assessment of counselling, and overall satisfaction but also long-term observation and patient reported outcome.”.
Minor editing of English language required
The manuscript has been re-checked by an English native speaker.
Reviewer 2 Report
Comments and Suggestions for Authors
The manuscript by Markowitsch et al., sought to investigate, through questionnaires, the practice of using amygdalin as unorthodox anticancer therapy among German physicians and healers. Although the proven toxicity of amygdalin and its inefficacy as an anticancer molecule have been known for over 40 years (PMID: 7033783), a survey on the behavior of medical care providers could nevertheless prove useful to document the persistent demand for this non-standard therapy. The manuscript language and the figure are excellent. The background information provided in the introduction and the presentation of methods and data, in my opinion, are acceptable, whereas the discussion is rather long.
Major criticism
Page 3, lines 110-111. Do the Authors have any information about the reasons that persuaded 22 contacted providers not to participate in the study?
Page 4, lines 153-154. Apparently, most patients taking amygdalin were also given conventional therapy. This means that medical care providers were unable to distinguish the separate effects of the two treatments, and their answers might have been influenced by this lack of knowledge.
Page 6, lines 227-230. On what basis did doctors and healers believed that patients would give a satisfaction rating of "very good" in three-quarters of cases, given that the question was addressed to them and not to the patients? Please be more clear.
Page 8, Table 11. Are the values ​​in grams of amygdalin expressed per square meter of body surface area?
Page 9, lines 296-302. One might ask, how did the participants concluded that negative side effects complained by their patients' were due to amygdalin and not to conventional therapy? Perhaps they disappeared when the dose of amygdalin was reduced?
Page 9, lines 303-311. Regarding cyanide, it is known that amygdalin has a narrow safety range and it was likely that at the highest doses this threshold would be exceeded. Why were there no questions aimed at capturing this information, especially since compliance was not checked?
Page 10, line 351. Were there no questions asked about the type of diet the patients followed? Amygdalin can generate cyanide through the action of β-glucosidase which is present in several foods including wheat, rice and certain types of fruit such as grapes, orange, melon, papaya, almonds and sweet cherry. Moreover, were the patients taking vitamins?
Minor remarks
Page 2, lines 58-59. Better use the capital letter to indicate the stereoisomer “6-β-D-glucoside”.
Page 3, line130. Why not use the exact p-value ? (for χ² = 5.37, the p-value is 0.020)
Author Response
Answers to the comments of referee 2
Comment 1: Page 3, lines 110-111. Do the Authors have any information about the reasons that persuaded 22 contacted providers not to participate in the study?
Our answer: We did re-contact those providers, who did not submit the questionnaire to us, although they had promised to do so. Several of them apologized for the delay and again promised to fill out the questionnaire. Some explained that they were momentarily too busy but would take care of the issue soon. In most cases, these promises were not kept and it may be assumed that there was no interest in participating in the survey. Whether these providers were indeed overloaded with work, whether they did not want to make their (anonymously submitted) data public or whether they feared negative impact in regard to the future use of amygdalin remains unclear. We have added this thought to the discussion (lines 613-616): “Whether these non-interested providers were actually overloaded with work, whether they did not want to make their (anonymously submitted) data public or whether they feared a negative impact on the future use of amygdalin remains unclear.”.
Comment 2: Page 4, lines 153-154. Apparently, most patients taking amygdalin were also given conventional therapy. This means that medical care providers were unable to distinguish the separate effects of the two treatments, and their answers might have been influenced by this lack of knowledge.
Our answer: This is correct. Particularly, table 17 should be interpreted in this context. The comment of the referee has now been considered in the discussion (lines 622-624): “Limitations of this study should be considered. Most of the patients taking amygdalin also received conventional therapy. Therefore, effects thought to be due to amygdalin (e.g. table 17) could have been due to conventional treatment.”.
Comment 3: Page 6, lines 227-230. On what basis did doctors and healers believed that patients would give a satisfaction rating of "very good" in three-quarters of cases, given that the question was addressed to them and not to the patients? Please be more clear.
Our answer: In fact, table 9 reflects the self-assessment of the amygdalin users. Not surprisingly, the response was “very good” in most cases. Being aware of this bias, we also included the question, how the amygdalin providers believe the patients would rank the information given. Here, the response pattern was more homogeneous. How far this classification was based on real patient opinion, or was just assessed by the provider is unclear. Indeed, to verify the responses given, it would be necessary to interview the patients themselves. However, due to data protection, we were not allowed to contact the patients of the respective healers and physicians. Still, we are planning a new study to evaluate tumor patients visiting healers or physicians for amygdalin infusion therapy. We have now added to the discussion (lines 624-626): “Some responses (e.g. table 8) particularly reflect the personal opinion of the participant and might be unduly positive. ”. We have also included (lines 626-629): “A new study is therefore aimed at attaining information from the patient’s point of view. It will include not only the patient’s assessment of amygdalin’s anti-tumor potential, assessment of counselling, and overall satisfaction but also long-term observation and patient reported outcome.”.
Comment 4: Page 8, Table 11. Are the values in grams of amygdalin expressed per square meter of body surface area?
Our answer: No, the values are not related to the body surface area. Although studies done by Moertel et al. (ref. 12, 32) were based on amygdalin given i.v. at 4.5 g/m2/day, further reports point to dosages independent of body surface area (ref. 34). As we noted in the discussion, amygdalin is generally ordered as 3 g vials for i.v. use. Therefore, the dosing was based on 3 g or a multiple thereof. We have now made this clear in the discussion (lines 502-503): “Dosage was not tied to the body surface area as practiced by Moertel et al. [12]. Rather, amygdalin was applied in 3 g units, i.e. 3 g or a multiple of 3 g.
Comment 5: Page 9, lines 296-302. One might ask, how did the participants concluded that negative side effects complained by their patients' were due to amygdalin and not to conventional therapy? Perhaps they disappeared when the dose of amygdalin was reduced?
Our answer: The participants were asked about negative side effects observed during or after amygdalin infusion. In this context, few side effects, typically indicating cyanide intoxication, were observed. One participant noted nausea which disappeared after 1 h, one participant noted signs of fatigue that disappeared after 4 h, and one patient complained of a mild headache, which disappeared when the infusion was slowed. These cases may point to amygdalin-related side effects. Still, negative effects due to the standard treatment regimen should not be ignored. In fact, side effects were also noted independent of amygdalin concentration or application time. We have now added to the discussion (lines 549-554): “It cannot be excluded that the (moderate) negative effects recorded in the present investigation were caused by the conventional therapy. However, this seems unlikely since patients were questioned about negative side effects observed during the actual amygdalin infusion. One participant noted nausea that disappeared after 1 h, another reported fatigue that disappeared after 4 h, and one patient complained of a mild headache that disappeared when the infusion was slowed. ”
Comment 6: Page 9, lines 303-311. Regarding cyanide, it is known that amygdalin has a narrow safety range and it was likely that at the highest doses this threshold would be exceeded. Why were there no questions aimed at capturing this information, especially since compliance was not checked?
Our answer: When we began the survey, we expected to obtain information on how providers would control amygdalin-tolerability. We were surprised that neither the HCN level nor compliance was being checked. Although we asked the providers why the HCN level was not being controlled, we did not ask about patient compliance. In retrospect, this aspect should have been investigated. We can speculate that the daily workload did not allow patient follow up. The level of interest must also be questioned, since the survey documents a lack of communication between the amygdalin providers and other healthcare providers, along with a limited interest in patient reported outcomes and physicians’ letters. We have now added to the discussion (lines 573-579): “The reason for the lack of compliance control can only be speculated upon. It is not clear whether amygdalin providers are aware of the risk associated with amygdalin. Possibly, their daily workload does not allow careful follow up. A low interest level might also be a reason why patients were not controlled at home. In fact, this survey documents a lack of communication between the amygdalin providers and other healthcare providers, along with a limited interest in patient reported outcomes and physicians’ letters. Due to the risk of overdosing …”
Comment 7: Page 10, line 351. Were there no questions asked about the type of diet the patients followed? Amygdalin can generate cyanide through the action of β-glucosidase which is present in several foods including wheat, rice and certain types of fruit such as grapes, orange, melon, papaya, almonds and sweet cherry. Moreover, were the patients taking vitamins?
Our answer: Nutrition is certainly a consideration when administering amydalin. It is correct that some fruits also contain β-glucosidase which may influence the efficacy of amygdalin when consumed together. It is also known that vitamin C can boost the influence of amygdalin when taken orally, thus increasing the risk of cyanide poisoning. We have dealt with this in an earlier article (Blaheta et al. Amygdalin, quackery or cure? Phytomedicine 2016,23,367-376). The present survey indicates that amygdalin treatment should be embedded into a multimodal concept including nutrition. Still, the patients enrolled in the current study were treated with amygdalin infusions, bypassing β-glucosidase driven metabolization. Since we have no information on self administered consumption of amygdalin at home, we believe this aspect of amygdalin treatment, as well as nutritional aspects, be reserved for a future study.
Minor remarks
Comment 1: Page 2, lines 58-59. Better use the capital letter to indicate the stereoisomer “6-β-D-glucoside”.
Our answer: It now reads “D-mandelonitrile-β-D-glucosido-6-β-D-glucoside”.
Comment 2: Page 3, line130. Why not use the exact p-value ? (for χ² = 5.37, the p-value is 0.020).
Our answer: This error has been corrected.